Warming increases the top-down effects and metabolism of a subtidal herbivore

Carr Lindsey A. 1 2 lacarr@email.unc.edu
Bruno John F. 1
1 Department of Biology, The University of North Carolina at Chapel Hill , Chapel Hill, NC , USA
2 Galápagos Science Center (a UNC-USFQ Joint Partnership) , San Cristobal Island, Galápagos Archipelago , Ecuador
Hawkins Stephen
Electronic publication date: 2013 Jul 25
Publication date: 2013
Volume: 1
Electronic Location ID: e109
Received 2013 Apr 10; Accepted 2013 Jul 2
Copyright: © 2013 Carr and Bruno
Copyright year: 2013
Copyright holder: Carr and Bruno
License: This is an open access article distributed under the terms of the Creative Commons Attribution License, which permits unrestricted use, distribution, and reproduction in any medium, provided the original author and source are credited.
License URL: https://creativecommons.org/licenses/by/3.0/

Keywords: Lytechinus, Metabolic theory of ecology, Galápagos, Ulva, Urchin, Herbivore, Algae, Temperature

Funding: Phycological Society of America Sigma Xi Grants-in-Aid of Research Wilson Memorial Fund This work was funded by the Phycological Society of America, Sigma Xi Grants-in-Aid of Research, and the Wilson Memorial Fund. The funders had no role in study design, data collection and analysis, decision to publish, or preparation of the manuscript.

==============================
Ecological theory and experiments indicate that warming can increase the relative strength of top-down effects via alterations to metabolic rates in several different systems, thereby resulting in decreased plant biomass at higher temperatures. However, the general influence of increased environmental temperature on top-down effects is not well understood in systems where organisms experience relatively large variation in temperature. Rapid ocean temperature changes are pervasive throughout the Galápagos Islands due to upwelling and downwelling of internal waves, ENSO events and seasonality. We measured the effect of large, but not uncommon, water temperature variation on the metabolism and grazing rate of a common subtidal herbivore and on photosynthesis of their algal prey in the Galápagos Islands in July 2012. We found that green urchin consumption and metabolism were greater at the higher temperature treatment (28°C), resulting in significantly less algal biomass. Our result that warming increased green urchin metabolic rates, even in a highly dynamic system, provides further support for a mechanistic link between environmental temperature and feeding rates. And further, our findings suggest individual response to temperature results in changes in top-down effects. And if this response is maintained over longer-time scales of days to weeks, this could translate to alterations of larger-scale ecological patterns, such as primary producer community composition and structure.

Introduction

The strength of herbivore-plant interactions determines the composition and distribution of primary producers in many marine communities (Lubchenco & Gaines, 1981; Hawkins & Hartnoll, 1983; Paine, 1992; Duffy & Hay, 2000; Burkepile & Hay, 2006). Several studies have found that this interaction is influenced by sublethal changes in environmental temperature, via alterations to metabolic rates (O’Connor, 2009; Kratina et al., 2012). For example, higher temperatures often cause increases in both primary production and consumption rates; however, due to differential temperature scaling of photosynthesis and respiration, consumption is predicted to increase relative to production at warmer temperatures (Allen, Gilloly & Brown, 2005; O’Connor, 2009; O’Connor et al., 2009). In some circumstances, this could lead to lower standing plant biomass due to stronger top-down effects.

A growing body of literature indicates that in a warming world the relative strength of top-down effects increases (within a non-lethal or non-stressful thermal environment), in freshwater, marine and terrestrial systems (O’Connor, 2009; Barton, Beckerman & Schmitz, 2009; O’Connor et al., 2009; Yvon-Durocher et al., 2010; Hoekman, 2010; Kratina et al., 2012). To date, such experimental studies have been conducted with relatively small but relevant changes in temperature (from ambient to + 6°C). It is unclear to what degree temperature influences species interactions in systems where organisms experience greater temperature variation. For example, it is possible that high natural temperature variability selects for physiological tolerance of temperature change (i.e., plasticity). Hoekman (2010) used the wide range of temperatures (10°C–35°C) experienced by the inquiline community in pitcher plants to determine how temperature influences the top-down effects of mosquito larvae on protozoa. He found mosquito larvae developed faster at the warmer temperatures, and consequently, had higher energy demands and fed on protozoa at a faster rate relative to slowly developing mosquito larvae. Yet there are few other studies that quantified the effect temperature has on top-down control in environments with highly dynamic temperature regimes.

We used the nearshore system in the Galápagos Islands to determine how temperature affects the metabolism and the strength of top-down effects of a common subtidal grazer, the green sea urchin (Lytechinus semituberculatus). Ocean temperature in the Galápagos is highly variable in space and time, ranging from 11°C to 31°C due to upwelling and downwelling of internal waves, El Niño-Southern Oscillation (ENSO) events and seasonality. Sea urchins are a key grazer on macroalgae in marine systems and can regulate the benthic algal community productivity and structure (Paine, 1980; Witman, 1985; Hereu et al., 2005; Brandt, Witman & Chiriboga, 2012). They are the most significant invertebrate grazer guild in the Galápagos Islands (Irving & Witman, 2009; Brandt, Witman & Chiriboga, 2012), and at high densities can convert macroalgal assemblages to urchin barrens or pavements of encrusting algae (Ruttenberg, 2001; Edgar et al., 2009). Therefore, if urchins exert strong top-down control in a system with large spatial and temporal variation in environmental temperature, warmer temperatures should strengthen the top-down effect of urchins on macroalgal assemblages and possibly result in increased urchin barrens.

We manipulated temperature in outdoor mesocosms and measured the effects on urchin grazing rate and metabolism, as well as plant photosynthesis. Our findings suggest warming strengthens the top-down effects of urchins and results in lower algal standing biomass.

Methods

Study system

The Galápagos Islands are located 965 km off the coast of Ecuador and are centered at the confluence of several oceanographic currents leading to a high degree of variability in ocean temperature and phytoplankton biomass. The Equatorial Undercurrent runs from west to east across the Pacific Basin and drives major upwelling as it collides with the western islands, resulting in cold and nutrient-rich waters around these islands (Houvenaghel, 1978; Houvenaghel, 1984). The cool Humboldt Current delivers nutrient-rich water to the southern edge of the archipelago (Kessler, 2006). The northeast region of the archipelago is strongly influenced by the warm, nutrient-poor waters of the North Equatorial Countercurrent (NECC, or the Panama Current) (Kessler, 2006). Both of these currents contribute to the South Equatorial Current (SEC), a westward flowing current that strongly influences the central region of the archipelago (Houvenaghel, 1984) (Fig. 1).

Figure 1 Map of Galápagos Archipelago and the surrounding currents.

Colored triangles relate to curves in Fig. 2. Map courtesy of A. Valdivia and modified from Schaeffer et al. (2008).

The dominant influence of the SEC changes seasonally, depending on the location of the Intertropical Convergence Zone (ITCZ) (Houvenaghel, 1984). The ITCZ is north of the equator during the Garùa (fine mist) season (May to December), and the Humboldt Current is the major contributor to the SEC. During the wet season (December to May) the ITCZ shifts towards the south and the dominant influence to the SEC is the NECC. This results in fluctuating gradients of temperature and resource availability throughout the central archipelago.

The maximum average SST across the archipelago occurs in February and March, with the minimum usually occurring in August or September (Houvenaghel, 1978; Schaeffer et al., 2008) (Fig. 2). Temperature data were collected from two sites on San Cristobal (the easternmost island in the archipelago). During 2011, there was a cold season low of 13°C in November 2011 and a high of 25°C in July 2011. In 2012, there was a warm season high of 30°C in February 2012 and a low of 20°C in May 2012 (Fig. 2 and File S1). This annual cycle is interrupted during ENSO events (Barber & Chavez, 1983; Chavez et al., 1999).

Figure 2 Daily water temperature (mean) measured in the shallow subtidal (<5 m) at Santiago, Isabela and San Cristobal.

The Galapagos Science Center is on San Cristobal. Water temperature measurements were recorded every 30 min with a HOBO temp logger. The smoothing curve was done in Kaleidagraph (v. 4.1.). A Stineman function was applied to the data. The output of this function has a geometric weight that is applied to the data points and ± 10% of the data range to generate the smoothed curve.

During the warm ENSO phase (El Niño), the easterly trade winds weaken resulting in a deeper EUC thermocline, warmer waters and decreased upwelling. This results in low standing stock of primary producers in the euphotic zone (Pennington et al., 2006); and, ultimately marine consumer populations, such as seabirds, marine iguanas, and sea lions, decrease in abundance (Valle et al., 1987; Laurie & Brown, 1990). In contrast, the cold ENSO phase (La Niña) occurs as the easterly trade winds strengthen, sea surface temperatures decrease, and upwelling intensifies resulting in higher standing stock of primary producers (Izumo, Picaut & Blanke, 2002). During La Niña, ocean temperature in the western islands can be as low as 11°C (Wellington, Strong & Merlen, 2001).

In addition to regional-scale spatiotemporal variability in temperature and resource availability, the upwelling and downwelling of internal waves result in extreme and rapid temperature changes over smaller spatial and temporal scales (Witman & Smith, 2003). For example, over a 53-week period on a rocky subtidal wall in the central archipelago at depths between 3 and 12 m, 20 cold water events were recorded where temperature dropped by 3–9°C over a 25 h period (Witman & Smith, 2003).

Study organisms

We measured the effect of temperature on the common subtidal herbivore Lytechinus semituberculatus (green sea urchin) and the green macroalga Ulva sp. Lytechinus, Tripneustes depressus (white sea urchin), and Eucidaris galapagensis (slate pencil urchin) are the three most common urchin species in the Galápagos Islands and together comprise 91% of urchin biomass (Brandt & Guarderas, 2002; Brandt, Witman & Chiriboga, 2012). In rocky subtidal habitats throughout the archipelago at depths between 1 and 5 m, Lytechinus and Eucidaris are the two most common urchins, while Tripneustes is rare (Table 1). Further, Lytechinus is a strong interactor and capable of converting algal turfs (brown filamentous turf, order Ectocarpales: Giffordia sp., Ectocarpus sp.) to urchin barrens in relatively short time periods, while Eucidaris does not have any detectable effect on the abundance of algal turfs (Irving & Witman, 2009).

Table 1 Natural variation in algal cover, temperature and urchin density in shallow subtidal habitats in the Galápagos Islands.

Sites were surveyed in June 2010 and July 2012. Site codes: LL = La Loberia (San Cristobal, SC), LT = Las Tijeretas (SC), PP = Punta Pitt (SC), CD = Cabo Douglas (Fernandina, FE), PE = Punta Espinosa (FE). Values represent means ± SE (n = 25 quadrats for urchin density and algal cover). Any other urchin species (e.g., Tripneustes) were present at densities less than 0.3 m−2 in all quadrats. Temperature was estimated with both in situ temperature loggers and satellite data (AQUA Modis) for the 30-day period prior to sampling, SE < 0.07 for all sampling points. Light intensity was measured using HOBO Light/Temperature Pendants before and during the experimental duration.

	2010	2012	
Site	LL	LT	PP	CD	LL	LT	PE	
Ulva cover (%)	9.7 ± 3.3	58.4 ± 3.5	2.9 ± 0.4	36.3 ± 5.1	7.8 ± 2.1	28.9 ± 2.7	23.4 ± 1.5	
Eucidaris density (m-2)	4.8 ± 1.3	1.6 ± 0.7	1.6 ± 0.2	8.4 ± 2.4	6 ± 1.9	2.1 ± 0.4	3.2 ± 1.3	
Lytechinus density (m-2)	10.2 ± 2.6	11.8 ± 1.8	11.2 ± 1.9	1.2 ± 0.7	14.9 ± 2	13.2 ± 1.6	17.6 ± 4.5	
Temperature (°C)	24.8	23.7	20.8	11.4	25.3	23.2	17.8	
Light intensity range
(lumens ft-2)	N/A	N/A	N/A	N/A	0-3102	0-2968	N/A	

Ulva sp. was chosen as a food item for Lytechinus for several reasons: (1) Ulva sp. are one of the most abundant macroalgal species, along with turf, crustose coralline algae, and Sargassum, in the Galápagos nearshore habitats (Vinueza et al., 2006; Vinueza, 2009); (2) ephemeral species, like Ulva, are highly palatable for herbivores (Carpenter, 1986); and (3) sea urchin fronts in the Galápagos appear to consume all macroalgal species except for brown species (e.g., Padina) (L. Carr personal observation) and damselfish turfs (Irving & Witman, 2009).

Urchins and Ulva were haphazardly collected from the southern part of San Cristobal Island (89°36′41.85″ W, 0°55′39.36″ S), and were immediately transported to the laboratory in buckets filled with seawater. All urchins were collected from a depth of ∼1.5 m. Study organisms were maintained in culture tanks indoors at 23°C (ambient sea water temperature) for two days prior to beginning water temperature adjustments. Assays were conducted in a shaded, outdoor facility at the joint UNC/USFQ Galápagos Science Center (San Cristobal Island, Galápagos).

Herbivore feeding rate

We conducted feeding rate assays in July 2012 to test the effect of two different temperatures (14°C or 28°C) on green urchin grazing rates on Ulva. Water temperature in the culture tanks was adjusted from ambient (23°C) to either 14°C or 28°C over a four-day period. This is within the time period of shallow subtidal temperature changes of this magnitude in the Galápagos (Witman & Smith, 2003).

Ulva and urchins were placed in 4-L plastic container mesocosms and received a fresh supply of temperature-conditioned seawater every 12 h. Temperature treatments were maintained with either Visi-Therm submersible individual heaters (Marineland, Blacksburg, Virginia, USA) or ice baths. Feeding assays were replicated twice (n = 5 replicates for each trial). Herbivore presence and absence treatments were randomly assigned in water tables. Each mesocosm was equipped with an iButton Thermochron datalogger (Dallas semiconductor, Dallas, Texas, USA) and water temperature was recorded every 5 min. Eight mesocosms were equipped with a HOBO Pendant temperature/light sensor (HOBO, Bourne, Massachusetts, USA) and relative light intensity was measured every 5 min.

Starting conditions for each mesocosm were 2.50 ± 0.004 g of wet mass Ulva tissue and either three urchins or no urchins (control to test for autogenic loss). The average test size for the urchins in the mesocosms was 3.55 ± 0.08 cm, which is representative of the green sea urchin populations in southern San Cristobal (n = 120 from two sites measured in May and June 2011: minimum 2.75 cm, maximum 5.8 cm. Mean ± 1 SE of 3.79 ± 0.61 cm).

Assays were terminated and final algal biomass was measured after 48 h (when ∼50% of algal tissue was consumed (Tomas et al., 2011)). Biomass consumption was estimated as ([Hi × Cf/Ci]−Hf), where Hi and Hf were the initial and final wet weights of algal tissue in the presence of herbivores, and Ci and Cf were initial and final wet weights of the controls. Relative light intensity levels did not vary between mesocosms and was 326.06 ± 19.7 lumens/ft2. These light levels are less than the average relative light levels at 1.5 m depth in the field (886.86 ± 74.07 lumens/ft2), but are within the range of light conditions experienced throughout the tidal cycle at the southern sites of San Cristobal Island during the month of June (Table 1).

Feeding rate assays were initially analyzed using a mixed model ANCOVA with one level of nesting. The analysis tested for one fixed effect (temperature treatment), covariate (urchin test size), and one random effect (temporal block). Consumption data were log transformed to meet the assumption of homogeneity of variances. The random effect was not significant (p = 0.183). Therefore, results were pooled and the random effect was dropped for final analysis of treatment effects. All statistical analyses were performed in R (v. 2.15.2).

Respiration and photosynthesis

To estimate the temperature response of metabolic pathways (net photosynthesis and respiration), we measured oxygen production and consumption rates for Ulva and green urchins in 0.6 L containers under conditions identical to the feeding rate assays. Initial and final oxygen concentrations were measured for Ulva (5 ± 0.08 g of leaf tissue) and paired blanks (seawater only) (n = 20 replicates) using a YSI-200 oxygen sensor (Yellow Springs Instruments, Yellow Springs, Ohio, USA). Samples of Ulva tissue (5 ± 0.08 g) were obtained by using three Ulva “rosettes” plucked from the substrate by the holdfast. Rosettes used were similar sizes and no cutting or tearing was necessary. After the initial measurement, aquaria were covered with plastic to minimize oxygen exchange with the air and left for 2 h. Net photosynthesis rates were estimated by subtracting measurements of dark oxygen consumption from light oxygen production.

Lytechinus oxygen consumption rates were measured according to methods described in Siikavuopio, Mortensen & Christansen (2008). Urchins were held at 23°C prior to oxygen consumption trials. One randomly selected urchin was placed into an airtight, closed-system respirometry chamber (0.6 L) and water temperature was gradually adjusted over a 10 h period to 14°C and then back up to 28°C over a 30 h period. Within the chamber was a mounted YSI 200 dissolved oxygen probe to measure oxygen concentrations (mg/L) and a pump for water circulation to prevent the development of strong oxygen and temperature gradients. The chamber was placed into a water bath to maintain temperature treatments. An individual urchin was then placed into the chamber and oxygen concentrations were measured every five minutes for one hour at each temperature treatment (14°C and 28°C). Trials were repeated for 11 urchins. The mean weight specific oxygen consumption rate (Q, mg O2 kg−1 h−1) was calculated with the equation of Karamushko & Christiansen (2002): Q=(C0−Ct)V/WT

C0 and Ct are the initial and final oxygen concentration (mg O2ߙl−1), respectively. V is the volume (l) of the chamber minus the test urchin volume (test urchin volume was estimated from their biomass). W is the biomass of the urchin in kg. T is the measurement time in hours.

Oxygen consumption and production test were analyzed with a t test on change in O2. All statistical analyses were conducted using R (version 2.15.2).

Results

Herbivore feeding rate

Temperatures in the cold mesocosms were maintained at 14.01 ± 0.08°C and 14.03 ± 0.07°C for trials 1 and 2, respectively. Warm mesocosms for trials 1 and 2 were 28.06 ± 0.09°C and 28.00 ± 0.04°C, respectively. The range of temperatures maintained across both trials was 13.54–14.47°C for the cold treatment and 27.56–28.5°C for the warm treatment (Fig. 3). Green urchin consumption was 46% higher at the warmer temperature (p < 0.0001) (Fig. 4A). Urchin test size was not a significant covariate (p = 0.87).

Figure 3 Mesocosm temperature values during both experiments.

Temperature in each mesocosm (n = 20 per temperature) was recorded every 5 min. with an iButton Thermochron datalogger (Dallas Semiconductor, Dallas, Texas, USA). The box corresponds to the 25th and 75th percentiles and the dark line inside the box represents the median consumption value. Error bars are the minimum and maximum.

Respiration and photosynthesis

Green urchin metabolism was significantly higher at 28°C than at 14°C (p < 0.001, Fig. 4B). Ulva oxygen consumption was greater at 28°C (at 14°C, 2.55 ± 0.11 g O2• g tissue-1• h-1; at 28°C, 3.24 ± 0.13 g O2• g tissue -1• h-1; p = 0.004). Oxygen production was also greater at 28°C (at 14°C, 3.88 ± 0.09 g O2• g tissue-1• h-1; at 28°C, 4.74 ± 0.15 g O2• g tissue -1• h-1; p = 0.01). However, net photosynthesis rates did not vary with temperature (p = 0.45, Fig. 4C).

Figure 4 Temperature effects on urchin grazing rates, metabolism and algal photosynthesis.

Temperature effects on (A) urchin consumption of Ulva, (B) urchin oxygen consumption, and (C) on Ulva net photosynthesis. Values are means ± SE; n = 10.

Discussion

Consistent with the predictions based on metabolic theory and a growing body of literature, our results indicated that sublethal warming significantly increases the strength of top-down effects. Specifically, we found a 14°C increase in temperature resulted in a 46% increase in grazing rate and lower standing plant biomass. Similar results have been found in other marine systems (O’Connor, 2009: with herbivores there was a nearly 100% decrease in algal net growth at high temperatures compared to growth at low temperatures with or without herbivores) and grasslands (Barton, Beckerman & Schmitz, 2009: warming of 1°C increased the strength of top-down indirect effects on grasses and forbs by 30–40%).

One limitation of our study was that the urchins and algae might have acclimated to the ∼5°C temperature change had we warmed the treatment tanks more slowly or maintained the experiment for longer. Thus, it is difficult to extrapolate to how slower or longer-term changes in temperature will affect urchin-algal interactions and, consequently, larger spatial scale changes in ecological patterns. However, the rate of temperature change during the acclimation period and experiment is similar to temporal patterns of temperature fluctuation experienced by urchins around San Cristobal and the Galápagos Archipelago in general (Palacios, 2004; Vinueza, 2009; Witman, Brandt & Smith, 2010). In this dynamic system, urchins rarely spend more than several days to a few weeks at the same temperature, suggesting that our experimental treatments were representative of the natural temperature regime.

While there was a significant temperature effect on consumer metabolism and feeding rates, there was not a significant temperature effect on algal photosynthesis rates following the 4-day acclimation period in this study. It is possible that this was because light, nutrients, carbon dioxide, or some other resource was limiting and thus warming could not stimulate photosynthesis. Further, the positive effect of increased temperature on algal photosynthetic rate can be reduced or reversed at sub-saturating light levels because warming can increase the light level needed to reach the compensation point (Davison, 1991). Light conditions in the experiment were within the range of light levels in the nearshore habitats in the Galápagos (Table 1) and further, most populations of subtidal algae are subject to subsaturating light conditions (Davison, 1991); therefore the experimental conditions likely reflect algal performance in the field. Our results are consistent with O’Connor (2009) which found no temperature effect for Sargassum with a 4°C temperature change.

Organisms throughout the nearshore habitats of the Galápagos Islands experience large and frequent temperature fluctuations (Witman & Smith, 2003; Palacios, 2004; Vinueza et al., 2006; Schaeffer et al., 2008; Vinueza, 2009; Witman, Brandt & Smith, 2010). Therefore, populations in this environment could be less metabolically sensitive to extreme temperature changes due to adaptation, resulting in metabolic responses that deviate from the predicted outcomes of metabolic theory. For example, in the rocky intertidal, adaptive regulation results in reduced snail metabolic rates at warmer temperatures (Marshall & McQuaid, 2011). However, our results provide evidence that in a dynamic system where organisms experience relatively large variation in environmental temperature, metabolism and consumption still scale with temperature, which is consistent with metabolic theory (Allen, Gilloly & Brown, 2005; O’Connor, 2009; O’Connor et al., 2009).

At the upper range of species’ thermal tolerances metabolism and consumption are predicted to scale differently and metabolic demands should outpace increased grazing intensity (Rall et al., 2010). Lemoine & Burkepile (2012) found Lytechinus variegatus consumption and metabolism scaled differently at temperatures beyond 29°C, a 9°C increase from the starting temperature treatment (20°C). Ultimately, urchin ingestion efficiency was decreased at the higher temperature, resulting in possible reduced consumer fitness. Because shallow subtidal temperatures around San Cristobal reach 30°C, future work should focus on understanding Lytechinus metabolism/consumption ratios at the highest temperatures in the Galápagos and the implications for species interactions and the strength of top-down effects under these conditions.

Our finding that warming increased urchin metabolism, even in a thermally variable system, provides further support for a mechanistic link between environmental temperature and feeding rates. Additionally, our findings indicate warming increases grazing intensity, which could in turn affect ecological patterns, such as primary producer community composition and biomass. For example, if these grazing rates are maintained over longer time-scales (i.e., days to weeks), warmer temperatures may increase the prevalence of urchin barrens in areas of the Galápagos with high densities of sea urchins because of the increased relative strength of top-down effects. This study focused on one algal species and thus we cannot extrapolate to how temperature would affect other algal-urchin interactions. Furthermore, macroalgal primary production is highly seasonal in the Galápagos nearshore habitats (Vinueza et al., 2006; Vinueza, 2009) and it is unclear what effect this could have on urchin-plant interaction strength. We also recognize that other factors influence urchin-plant interactions and primary production. Future work should focus on elucidating the stability of this interaction under larger temporal and spatial scales, and various environmental conditions (i.e., different seasons, ENSO cycles, varying light intensities, etc.) and with different algal species.

The absence of macroalgae in intertidal and shallow subtidal habitats during warm periods (e.g., El Niño) in the Galápagos Islands is generally attributed to the decreased strength of bottom-up forcing (i.e., upwelling) and subsequent lack of nutrients (Vinueza et al., 2006). However, temperature stress (i.e., desiccation) does play a minor role in regulating intertidal algal biomass in the Galápagos (Vinueza, 2009). Therefore, future work should focus on understanding the constraints of physical stress on macroalgal growth in the Galápagos Islands, as stress is known to alter the relative importance of bottom-up and top-down effects (Thompson, Norton & Hawkins, 2004).

Low macroalgal biomass in warm seasons and years could also be due in part to increased grazing intensity due to higher temperature (e.g., O’Connor et al., 2009). Both mechanisms could be operating (i.e., changes in top-down and bottom-up control), although the relative strength of these mechanisms in influencing large-scale ecological patterns in upwelling systems is unknown. In freshwater pond systems, Kratina et al. (2012) found a negative interaction between warming and nutrient input on total phytoplankton biomass, suggesting a shift toward stronger top-down and weaker bottom-up effects with warming regardless of nutrient availability. However, the mechanism behind this shift is not known. Therefore, future warming could result in stronger consumer control in systems where nutrients are plentiful (i.e., upwelling or well-mixed aquatic systems) that could lead to short- and/or long-term changes in community structure and function.

Supplemental Information

File S1 Urchin oxygen consumption values

Click here for additional data file.

File S2 Field temperature raw data for Santiago, Isabela and San Cristobal

Click here for additional data file.

File S3 Mesocosm temperature values

Temperature was recorded every 5 min and averaged over the 48 h period.

Click here for additional data file.

File S4 Algal oxygen production and consumption values

Click here for additional data file.

File S5 Algal weights before and after experimental duration (48 h)

Click here for additional data file.

We thank the Galápagos National Park, S Walsh, C Mena, P Page, L Vaca, and the Galápagos Science Center for providing field and logistical assistance. I Vu provided valuable field assistance. We thank J Witman whose input improved the experimental design. Various stages of this manuscript benefited from comments by R Gittman, J Fodrie, C Cox, A Valdivia, E Darling and four reviewers.

Additional Information and Declarations

Competing Interests

Author Contributions

Field Study Permissions

John Bruno is an Academic Editor for PeerJ. There are no other competing interests.

Lindsey A. Carr conceived and designed the experiments, performed the experiments, analyzed the data, contributed reagents/materials/analysis tools, wrote the paper.

John F. Bruno conceived and designed the experiments, analyzed the data, contributed reagents/materials/analysis tools, wrote the paper.

The following information was supplied relating to ethical approvals (i.e., approving body and any reference numbers):

Galápagos National Park, Santa Cruz Island, Galápagos Islands, Ecuador, permit #PC-17-12.

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
