# Peer review of "Warming increases the top-down effects and metabolism of a subtidal herbivore"

_PeerJ, doi:10.7717/peerj.109_

## Round 0.1 · original submission · Major Revisions

We have received two contradictory reviews.

I concur more with the positive reviewer, but the more negative reviewer has some potentially valid points. Therefore I would like you to address the suggestions of the positive reviewer in your resubmission.

The negative reviewer has some theoretical reservations that need to at least be recognized in the Introduction. There is also methodological concerns about the ratio of algal material to volume of water and interactions of light and temperature in photosynthesis. These should be addressed in the ms or refuted.

Please either adjust your ms to take on board these negative comments or refute them in a covering letter with the resubmitted manuscript. I then intend to send this to another reviewer for a fresh view.

With these concerns addressed, a resubmitted MS is likely to be favourably received by additional review.

Reviewer 1 ·

Basic reporting

This study tests the relative effects of two temperature treatments (14°C and 28°C) on the relative feeding rates and metabolism of an urchin as compared to net productivity of an alga. The central idea being tests is that there are often relatively rapid changes in water temperature, and so these tests examine the effect of these changes on the strength of top down control.

The rationale given for the study is that some studies have suggested that increasing temperature will increase top down control. This is based on the assumption that as temperature increases, so will the metabolic demand of herbivores, and so they should compensate for increased demand by eating more. Unfortunately, this is not always true. This study falls into the unfortunately common trap that many “Metabolic Theory of Ecology” studies seem to frequently stumble into, namely that increasing temperatures will generally increase rates of metabolism.
Physiological studies in terrestrial environments have shown that often organisms live in conditions just below their thermal optimum (or, at least they do near the center of their range). Thus, slight increases in temperature should increase performance to some optimum (but see contrasting theories by authors such as Angilleta). The problem is, above that optimum, performance begins to drop. This has been shown for many species. For example, the sprint speed of some lizards increases to an optimum, then declines above that threshold. Ditto with rates of foraging. Similar studies have been conducted on plants, showing that net productivity increases with temperature to some optimum, and then decreases. In other words, it all depends on the starting point: below the optimum, increasing temperature will increase performance, above that optimum increasing temperature will decrease performance. This is ignored by MTE, and in cases where it has been shown to work, it is (as the authors indicate) under small changes in temperature where organisms are probably being pushed to their optimum.

Experimental design

So, the use of two temperatures is difficult to interpret, as are any potential implications for top down control. Since these are short term temperature excursions, their effects will be temporary as well. Under longer-term changes in mean temperature, acclimation will likely play a significant role.

I am also rather skeptical that the photosynthesis and respiration rates of 5g of alga can be measured in a 4L container. No mention of how precisely 5g was obtained, so I assume the alga were cut prior to experiments, which causes all sorts of problems. If they were not cut, then they should have been weighed and P and R normalized to a per g basis. Finally, it is possible that the effects of temperature covary with the effects of light. Since experiments were conducted under one light condition, the results are again hard to resolve.

Validity of the findings

Overall, this strikes me as an interesting set of preliminary measurements conducted over a period of a month at a remote field station, but they are not yet ready for prime time.

·

Basic reporting

See below

Experimental design

See below

Validity of the findings

See below

Additional comments

Review of “warming increases the top-down effects and metabolism of a subtidal herbivore” by Carr and Bruno.

General comment:

The manuscript describes an interesting study on the effect of warming on a herbivore-plant interaction. The paper is very well written, the rationale and methods are sound and, in general, the data are appropriately treated and interpreted. The study region is of great interest, due to the pronounced environmental variability that characterises the Galapagos, and the paper will be a worthwhile addition to the marine climate change ecology literature. I have 2 major points and some minor criticisms that may improve the manuscript.

Major concerns:

The authors make some fairly far-reaching conclusions concerning urchin barrens and climate variability based on 1 (albeit replicated) experiment on 1 herbivore and 1 algal species. As such, I feel that certain limitations of the study could be addressed, or at least discussed, and perhaps some of the inferences should be toned down.

1. The experiment was conducted within a single season, at a single location. Seaweed physiology can be highly seasonal and responses including primary production can vary dramatically in space and time (e.g. Delebecq 2013 marine biology). Although ulva spp. are typically opportunistic and exhibit less seasonality, it would be useful to examine temporal variability in herbivore-plant interaction strength. I wondered if the authors have any additional data (from pilot experiments, even with a less-than-optimal experimental design) that could be included to provide indication of the stability of the interaction? At the least, this should be mentioned as a caveat in the discussion.

2. Similar to point #1, the experiments included a single algal species. This raises some questions relating to how transposable the results are to ‘real’ conditions influencing the formation of urchin barrens in the field. For example, what is the green sea urchin’s preferred diet in the field? Does ulva spp. constitute a major diet item? What are the dominant algal species in these habitats and might they have greater ability to compensate/acclimatise to short term temperature variability than ulva? Again, if the authors have any additional data this could be incorporated or this could be discussed.

Minor points:

Introduction

Line 21: I would suggest that Yvon-Durocher et al, 2010, Phil Trans Roy Soc B would be a useful reference here.

Methods
Line 42: I think a map figure is needed, which should illustrate the influence of the major currents on the archipelago, as well as the locations of the ‘temperature’ islands in figure 1 and the main study location.

Line 83: Information is provided on the dominant urchin herbivores but not on benthic algal species. How abundant (relatively) is ulva spp? How many species make up the turf assemblage and which are the most important?

Line 93: Annual temperature profiles are provided for 2 islands (Fig 1) but neither of which are the main study island (only temperatures at San Cristobal for the period immediately before field surveys are provided). What is the thermal regime like at the study island? Can the authors present in situ temperature data (as in Fig 1) for San Cristobal in addition to the 2 islands already shown? If not, is the thermal regime similar to either of these islands (again, this would be easier to decipher if a map was provided to show the proximity of the islands).

Line 115: Average test size of 3.55cm seems quite small – were these all adult urchins or a mixture of juvenile and adults? Should be stated.

Discussion

Line 198: ‘species’ or ‘population’ or both? Presumably many of these species are found in more stable regions so only populations have (potentially) adapted to environmental variability.

Line 203: Marshall and McQuaid citation not in reference list

Line 224: Parentheses around citation need correcting

Line 239: Need citations to support the statement

Figure 1: The date format on the x-axis could be reformatted to month (i.e. JASONDJFMAMJ) with the year below to improve clarity.

Figure 2: Is this both experiments combined? State n in the legend.

Figure 3: State n in the legend.

---

## Round 0.2 · Minor Revisions

These final comments are essentially minor. I have also had the paper independently reviewed by an associate in my research group. I have also refereed the resubmitted ms myself. I think the paper is much improved and now reads well. Both my associate and I feel that a bit more still needs to be made of the difficulties in extrapolating short term studies in the discussion. We suggest moving paragraph L. 237-243 much earlier into the discussion, maybe after L. 212

Some minor points that need addressing and suggestions include:

L. 15 Change “some” to “many marine”
L. 15 Add the reference Hawkins & Hartnoll 1983 (OMBAR) to the reference list
L. 18 Add reference Thompson et al. 2004 (Ecology) and change sentence to: “…(O’Connor 2009, Kratina et al. 2012) or stress (Thompson et al. 2004).”
L.23-24 Change to “A growing body of literature demonstrates that in a warming world the relative strength of top-down effects increases in freshwater...”
L. 41 Give the full name of ENSO here (not in L. 78)
L. 41-46 Change sentences to avoid repetition in wording used. We suggest changing to:
“..…Hereu et al. 2005, Brandt et al. 2012), the most significant invertebrate grazer guild in the Galapagos Islands (Irving and Witman 2009, Brandt et al. 2012). At high densities they can convert…”
L. 46 Change “are a strong interactor” to “exert strong top-down control”
L. 48 Remove “theory predicts”. Change “will” to “should”
L. 66 Change to “… the SEC changes seasonally, depending on…”
L. 72-73 Change to “…across the archipelago occurs in February and March, with the minimum usually occurring in August of September…”
L. 74-77 Change sentence structure. We suggest:
“Temperature data were collected from two sites on San Cristobal (the easternmost island in the archipelago). During 2011 and 2012, there was a low of 13®C in November 201 and a high of 25®C in July 2011. In 2012, there was a warm season high of…”
L. 78 Remove full name after ENSO (but mention in L. 41 instead, see above).
L. 78-79 Remove everything after “interrupted during ENSO events”, but leaving references in
L. 83 Remove “and” at the beginning of the sentence and use semicolon to connect both clauses.
L. 98 remove “hereafter referred to by genus”
L. 108 change “prey item” to “food item”
L. 132 remove “as a temporal block”
L. 134 Remove semi-colon after (Dallas semiconductor, Dallas, Texas, USA)
L. 140-142 We suggest to remove the sentence beginning “Green sea urchin test sizes...” but add the following information in brackets after “San Cristobal”:
“(n = 120 from two sites measured in May and June 2011: minimum 2.75 cm, maximum 5.8 cm. Mean ± 1 SE of 3.79 ± 0.61)”
L. 164 Change to “Samples of Ulva tissue (5 g ± 0.08 g) were obtained by using three…”
L. 171, L. 180-181, L. 223, L. 274-275 The brackets should only include the year of the publications
L. 196 Add by how much consumption rates were greater at higher temperature, in %
L. 206 it should be “indicated” instead of “indicate”
L. 207 this percentage value (46%) should be referred to previously in the results section
L. 208 it should be “have been found” instead of “were found”
L. 208-212 Sentence structure. We suggest the following:
“Similar results have been found in other marine systems (O’Connor 2009: with herbivores, there was a nearly 100% decrease in algal net growth at high temperatures compared to growth at low temperatures with or without herbivores) and grasslands (Barton et al. 2009: warming of 1®C increased the strength of … by 30-40%).”
L. 237-243 As mentioned before, we believe this should be addressed earlier in the discussion. 4 days is a very short time period. Extrapolation to long-term changes is difficult
L. 257 We suggest you could refer to the following reference in brackets after “structure”: (see also work by Jenkins et al. 2001, MEPS 211, 193-203, on comparisons of limpet grazing intensity with latitude)
L. 264 Should it be “besides” instead of “as well as”?
L. 269 Remove the comma after Galapagos Islands
L. 274 We suggest you could add something like the following sentence:
“Thompson et al. (2004) suggested that on rocky shores temperature stresses acting temporally (e.g. season) and spatially could modify the relative importance of top-down control and bottom-up forcing.

---

## Round 0.3 · accepted · Accept

Thanks for the revisions.